# Insecticide-treated house screening protects against Zika-infected *Aedes aegypti* in Merida, Mexico

**Pablo Manrique-Saide**[1]*, **Josué Herrera-Bojórquez**[1], **Anuar Medina-Barreiro**[1], **Emilio Trujillo-Peña**[1], **Josué Villegas-Chim**[1], **Nina Valadez-González**[2], **Ahmed M. M. Ahmed**[1,3], **Hugo Delfín-González**[1], **Jorge Palacio-Vargas**[4], **Azael Che-Mendoza**[1], **Norma Pavía-Ruz**[2], **Adriana E. Flores**[5], **Gonzalo Vazquez-Prokopec**[6]

1 Unidad Colaborativa para Bioensayos Entomologicos, Campus de Ciencias Biologicas y Agropecuarias, Universidad Autonoma de Yucatan, Merida, Yucatan, Mexico, 2 Centro de Investigaciones Regionales, Unidad Biomédicas, Universidad Autonoma de Yucatan, Merida, Yucatan, Mexico, 3 Faculty of Agriculture, Assiut University, Assiut, Egypt, 4 Servicios de Salud de Yucatan, Merida, Yucatan, Mexico, 5 Facultad de Ciencias Biologicas, Universidad Autonoma de Nuevo Leon, San Nicolas de los Garza, Nuevo Leon, Mexico, 6 Department of Environmental Sciences, Emory University, Atlanta, Georgia, United States of America

* pablo_manrique2000@hotmail.com

**Data Availability Statement:** All relevant data are within the manuscript and Supporting Information files.

## Abstract

### Background

The integration of house-screening and long-lasting insecticidal nets, known as insecticide-treated screening (ITS), can provide simple, safe, and low-tech *Aedes aegypti* control. Cluster randomised controlled trials in two endemic localities for *Ae. aegypti* of south Mexico, showed that ITS conferred both, immediate and sustained (~2 yr) impact on indoor-female *Ae. aegypti* infestations. Such encouraging results require further validation with studies quantifying more epidemiologically-related endpoints, including arbovirus infection in *Ae. aegypti*. We evaluated the efficacy of protecting houses with ITS on *Ae. aegypti* infestation and arbovirus infection during a Zika outbreak in Merida, Yucatan, Mexico.

### Methodology/Principal findings

A two-arm cluster-randomised controlled trial evaluated the entomological efficacy of ITS compared to the absence of ITS (with both arms able to receive routine arbovirus vector control) in the neighbourhood Juan Pablo II of Merida. Cross-sectional entomological surveys quantified indoor adult mosquito infestation and arbovirus infection at baseline (pre-ITS installation) and throughout two post-intervention (PI) surveys spaced at 6-month intervals corresponding to dry/rainy seasons over one year (2016–2017). Household-surveys assessed the social reception of the intervention. Houses with ITS were 79–85% less infested with *Aedes* females than control houses up to one-year PI. A similar significant trend was observed for blood-fed *Ae. aegypti* females (76–82%). Houses with ITS had significantly less infected female *Ae. aegypti* than controls during the peak of the epidemic (OR = 0.15, 95%CI: 0.08–0.29), an effect that was significant up to a year PI (OR = 0.24, 0.15–0.39). Communities strongly accepted the intervention, due to its perceived mode of action,

**Funding:** Research funding was provided by the Canadian Institutes of Health Research (CIHR) and IDRC (Preventing Zika disease with novel vector control approaches, Project 108412) to PMS. The funders had no role in study design, data collection and analysis, decision to publish, or preparation of the manuscript.

**Competing interests:** The authors have declared that no competing interests exist.

the prevalent risk for *Aedes*-borne diseases in the area, and the positive feedback from neighbours receiving ITS.

## Conclusions/Significance

We show evidence of the protective efficacy of ITS against an arboviral disease of major relevance, and discuss the relevance of our findings for intervention adoption.

### Author summary

We evaluated the efficacy of protecting houses with insecticide-treated nets permanently fixed with aluminium frames on external doors and windows on *Ae. aegypti* infestation and arbovirus infection during a Zika outbreak in Merida, Yucatan, Mexico. Houses protected with screens were ≈80% less infested with *Aedes* females and very importantly, had significantly less infected female *Ae. aegypti* during the peak of the epidemic. Communities strongly accepted the intervention, due to its perceived mode of action, the prevalent risk for *Aedes*-borne diseases in the area, and the positive feedback from neighbours. House screening provides a simple, affordable sustainable method to reduce human-vector contact inside houses and can protect against dengue, chikungunya and Zika.

## Introduction

The modification of human housing to make it refractory to insect vectors is gaining renewed impulse as a new paradigm for mosquito control [1,2]. Particularly, the use of mosquito-netting (mesh) as a physical barrier to prevent mosquito entry has been found protective against malaria and dengue in some observational studies [3,4]. Noteworthy, recent evidence from field trials on house-screening (HS) conducted primarily in Africa have shown significant protection against malaria [3,5–8] while being widely accepted by communities [5,9].

The principle of "keeping the vector out" is at the core of effective housing interventions to sustainably prevent vector-borne diseases and it is currently encouraged by the World Health Organization [1,10]; yet, it has been largely ignored for policies & programs for the prevention and control of *Aedes*-transmitted diseases (ATDs). In 2017, a research-to-policy forum convened by TDR/WHO [11], finally identified HS as a promising vector management approach for the prevention and control of ATDs. However, the need on stronger epidemiological evidence was also recognised [11,12]. HS is not included in the current WHO dengue guidelines [13] but, given its potential and wide-ranging benefits, it is a strong candidate for further trials to evaluate its effectiveness and optimal delivery within an Integrated Vector Management (IVM) framework that may include social mobilization and collaboration within the health sector and beyond [14].

The integration of HS and Long-lasting insecticidal nets (LLIN), known as insecticide-treated screening (ITS) [15], can provide simple, safe, and low-tech *Aedes* control. Projects supported by TDR/IDRC within the "Eco-Bio-social Research" and "Ecohealth" programmes in Mexico showed that LLIN affixed as ITS on doors and windows act as a physical/chemical barrier [16] and confer sustained protection for indoor-female *Aedes aegypti* [17–19]. Cluster randomised controlled trials in two endemic localities for *Ae. aegypti* and ATDs of south Mexico, showed that ITS conferred both, immediate and sustained (~2 yr) impact on indoor-female *Ae. aegypti* infestations, even in the presence of locally high pyrethroid resistance. In

the communities where it was implemented, ITS was considered a sustainable, popular and easy to adopt intervention [20], with a significant effect on indoor *Ae. aegypti* and therefore human-vector contacts. Such encouraging results require further validation with studies quantifying more epidemiologically-related endpoints, including ATD infection in *Ae. aegypti*.

Under the support of the International Development Research Centre Government of Canada (IDRC) we evaluated the community acceptance and efficacy of ITS on *Aedes aegypti* infestation and arbovirus infection during a Zika outbreak in Merida, Yucatan, Mexico. Capitalizing on the novel introduction of Zika virus (ZIKV) into Merida [21], we quantified the relative efficacy of ITS in comparison to the absence of ITS in the context of continued routine vector control reactive to the report of symptomatic ZIKV cases.

## Methods

### Ethics statement

This study received clearance from the ethical committee of the Ministry of Health of Yucatan. Written informed consent was obtained for each participating household (householder over the age of 18) in the beginning of the study.

### Study site

The study was developed in the area known as "Juan Pablo II" (~ 3.95 km$^2$ which includes the neighbourhoods Juan Pablo II, Juan Pablo II Segunda etapa and Ampliacion Juan Pablo II) within the city of Merida in the Mexican state of Yucatan, South Mexico (Fig 1). The average altitude of site is nine meters above sea level. Climate is mainly warm with an annual average temperature of 26˚-27˚C (36˚C max- 18˚C min). Two seasons can be clearly distinguished: a rainy season, in May to October (with most of the rainfall from June-October) and a dry season from November to April. The rainy season is associated the dengue risk season (transmission increases 80% approximately, although there is continuous transmission throughout the year) and marks the starting point for major vector control activities.

Merida, capital and major urban centre of the state of Yucatan, has a population of 814,435 people living in 272,418 households [22]. In the national context, Merida is one of the cities that reported the highest proportion of dengue cases in the last 18 years [23] and has accounted for ≈50% of all dengue cases in Yucatan during the last decade. The first cases of chikungunya in Merida and a subsequent outbreak (1,669 cases) occurred in 2015 and transmission decreased in the following years (11 cases in 2016, and 0 cases in 2017–2018) [21]. Zika transmission was detected in May 2016 reporting in the end of the year 2,199 cases; the transmission decreased to 24 cases in 2017, and 28 cases in 2018 [21]. Juan Pablo II has approximately 4,100 households, and with > 20,000 inhabitants is one of the most populated neighbourhoods in the city. Juan Pablo II was selected in consensus with the local Ministry of Health, because epidemiologically is considered the second neighbourhood most important for the local dengue control programme (from 2011–2018 it concentrated 5.4% of all dengue cases reported in Merida).

### Study design

The study followed a two-arm cluster-randomised controlled trial design, comparing five clusters with the intervention versus another five without ITS as control for one year, as in previous studies [17–19]. An area (0.24 km$^2$ comprising 31 blocks and 1,038 houses) was divided in ten clusters (nine clusters of three blocks and one of four blocks) that were randomized to receive the intervention or to remain as controls (Fig 1). We powered the study to detect a

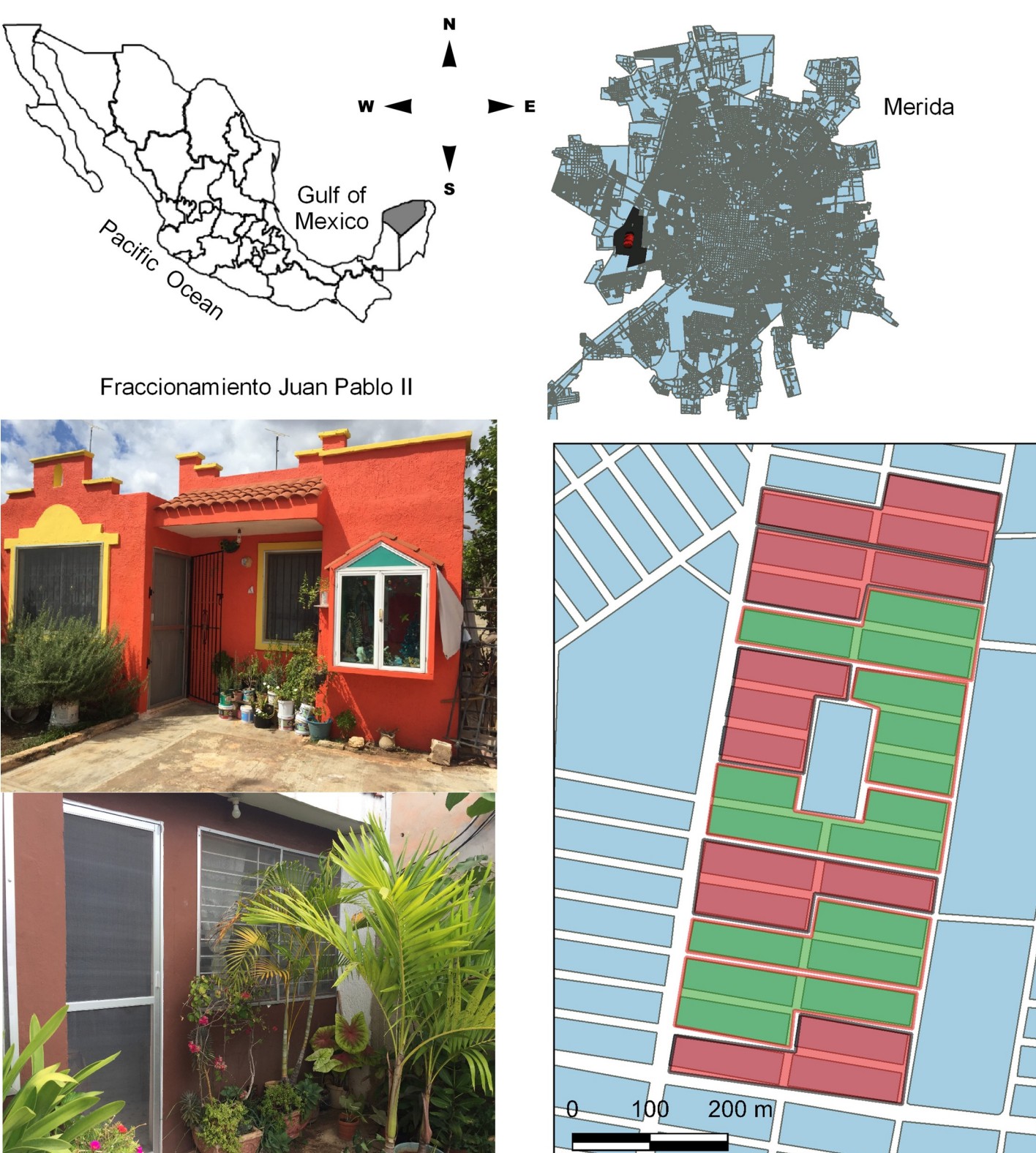

**Fig 1. Study site.** The city of Merida, Yucatan, Mexico and the location of the neighbourhood Juan Pablo II. Intervention clusters are shown in green and control clusters are coloured in red. Photographs show *Aedes aegypti* proof-houses with insecticide-treated screens mounted on aluminium frames and fixed to external doors and windows of treated houses.

significant difference in our primary entomological endpoint: the density of *Ae aegypti* indoors (collected after a 10-min Prokopack aspiration session). Based on an expected effect size of 70% in the reduction of *Ae. aegypti* indoors by ITS [18] from an expected mean baseline number of 4.4 ± 9 [24], an alpha of 0.05 and a power of 80%, we estimated a total of 134 houses per arm (268 total houses) to detect a significant difference between groups (https://clincalc.com/stats/samplesize.aspx).Therefore, our population of over a thousand houses provided enough statistical power to evaluate a difference at even a lower effect size than 60%. The implementation of the intervention (installation of ITS, see below) was carried out during June-July 2016. The intervention was evaluated with entomological indicators of impact e.g. female *Aedes*, blood fed female *Aedes* and female *Aedes* infected with any ATD. It is worth mentioning that the study was planned and implemented prior to ZIKV introduction in the study site and took advantage of the opportunity to compare the impact before and after the introduction of ZIKV. Originally, the primary goal was to evaluate de entomological efficacy (presence and abundance of the vector). Infection of mosquitoes with arboviruses was already planned but was originally a secondary goal since we cannot predict when and where an outbreak can occur.

Both areas received routine vector control, which in Merida occurs in response to reported symptomatic ATD cases and elevated entomological indices [25]. These activities included: outdoor-spraying with organophosphates (chlorpyrifos-ethyl, malathion), indoor spraying with carbamates (propoxur, and bendiocarb) and a pyrethroid (deltamethrin) and larviciding with temephos, novaluron and spinosad.

## Insecticide-treated house screening

As described in previous studies [17–19], Duranet long-lasting insecticidal nets material (0.55% w.w. alpha-cypermethrin-treated non-flammable polyethylene netting [145 denier; mesh1/4132 holes/sq. inch]) was mounted in aluminium frames custom-fitted to doors and windows of houses in collaboration with a local small business (Fig 1).

A total of 420 households which were suitable for installation, inhabited and that agreed to participate (from an expected number of 500 houses) from intervention clusters (84% of coverage) were protected with ITS. An average (mean ± standard deviation) of two doors (1.8±0.31) and six windows (6.24±1.32) by house were installed in each intervention cluster. During the installation, at least one person in every household received information from research staff about the proper use and maintenance of ITS [26]. The total average cost of the ITS (materials and professional installation) was US $147.06 per house.

## Vector and arbovirus surveillance

**Entomological field studies.** Indoor adult mosquito collections were performed as in previous studies [17–19], in a randomly selected sub-sample of 30 houses from each cluster (n = 150 houses per arm). Three cross-sectional entomological surveys were conducted in intervention and control clusters. The baseline survey was completed in May 2016 (dry season) and was followed by post-intervention (PI) surveys over 2016–2017 during the dry (low vector abundance) and wet (high vector abundance) subsequent seasons. Indoor adult mosquitoes were collected with Prokopack aspirators [27] for a 15-min period per house. Collections within each cluster were performed on the same day between 09:00–12:00 hrs. by 3 teams of 2 skilled collectors each. All mosquitoes collected were identified to species and sex.

**Presence of virus in mosquitoes.** The study included the detection of dengue (DENV), chikungunya (CHIKV) and Zika (ZIKV) viruses in female *Ae. aegypti* collected in the entomological surveys. After identification, female *Ae. aegypti* were vialed in pools of 1–9 individuals

for each condition (blood fed, and non-blood fed) in RNAlater and transported to the Haematology Laboratory of the Regional Research Center at the Autonomous University of Yucatan (CIR-UADY) for analysis. The total sample for virus testing was 103 pools totalling 161 blood-fed females and 36 pools totalling 53 non-blood fed females. Laboratory diagnostics was blinded to the cluster allocation.

RNA extraction from mosquito pools was conducted using the manual extraction protocol [28] followed by confirmation of yield and purity of the RNA using a spectrophotometer (Nanodrop's AB equipment). After extraction, molecular detection of ZIKV in mosquitoes was performed with the use of the primers and probes reported by [29]. For detection and differentiation of RNA from CHIKV and DENV we used primers and probes from the Centers for Disease Control and Prevention (CDC; catalog # KT0166). The rRT-PCR [27] was done with the QIAGEN OneStep RT-PCR Kit (QIAGEN catalog 210212). To validate our RT-PCR results, we used the tissue culture supernatant of infected Vero cells heat inactivated of ZIKV strain Puerto Rico 2015, CHIKV strain Puerto Rico 2013 and tissue culture supernatant of infected mosquito-derived C6/36 cells heat inactivated for DENV type 1 (DENV-1) strain Puerto Rico 1998, for DENV-2 strain Puerto Rico 1998, for DENV-3 strain Puerto Rico 2004 and for DENV4 strain Puerto Rico 1998. The results are expressed as CT values that are inversely proportional to the viral RNA concentration in each sample. CT values were determined based on positive and negative controls, and CT values below 38 cycles were considered positive.

## Social assessment of the intervention

As in previous studies on ITS in Mexico [20,25], the team performed a social assessment focused on communities' acceptances and their perceived efficacy about the intervention. Household-surveys were applied to 140 families randomly selected within intervention clusters to address the social reception of the project six months after the interventions was installed. Topics considered were: acceptance of intervention, opinions on the installation process, perception of temperature increase associated to screenings material, satisfaction in the reduction of mosquitoes inside houses, perception on positive cases of DENV/CHIKV/ZIKV reported by the families after the installation of ITS, and recommendations for scaling-up ITS-method.

## Data analysis

From indoor Prokopack adult collections we calculated: a) Houses positive (presence of at least one) by female *Aedes* (%), b) Houses positive by blood fed female *Aedes* (%), c) Number of female *Aedes* per house, and d) Number of total blood fed *Aedes* per house. We also report the prevalence of positive houses to indoor-female *Aedes* with arbovirus infection (houses positive to *Aedes* females/house with at least one pool positive to arbovirus). Logistic regression models (for presence-absence mosquito data) and negative binomial models (for count data) accounting for each house's cluster (cluster-robust SE calculation) were performed for each cross-sectional entomological evaluation survey. Odds ratios (OR) and incidence rate ratios (IRR) with 95% CIs were assessed and significance expressed at the 5% level. Analyses were performed using STATA 13.0 (Stata Corp, College Station, TX, USA), and graphics were done in R (https://www.r-project.org). Such values from the infection calculation were used to calculate a measure of epidemiological efficacy, as $ITS_{eff} = (1- OR) \times 100$ [30]. This value, which ranks between 0 and 100, indicates the proportional reduction in *Ae. aegypti* infection in treatment arms, in comparison to control arms.

## Results

### Impact of ITS on indoor adult mosquitoes

A total of 613 adult mosquitoes were collected resting inside the houses of Merida during the whole study period. *Ae. aegypti* was the most abundant (75.5%, 249♂, 214♀) mosquito species, followed by *Culex quinquefasciatus* (23%, 69♂, 72♀), a few *Cx. nigripalpus* (0.8%, 2♂, 3♀), and *Ochlerotatus taeniorhynchus* (0.6%, 4♀). Most of the specimens were collected during the rainy season in October 2016 (76.9%).

Adult *Ae. aegypti* indoor entomological indicators were calculated at baseline (dry season 2016), and after six (wet season 2016) to twelve (dry season 2017) months post-ITS intervention (Table 1 and Fig 2). At baseline, statistically similar infestation levels were quantified in both study arms. After the installation of ITS (wet season, 6 months PI survey), significant differences between treatment and control arms were observed on the positivity (presence) of adult females (OR = 0.15, 95% CI 0.081–0.26, P<0.001) and blood fed females (OR = 0.18, 95% CI 0.097–0.325, P = <0.001). The statistical difference between treatment and control arms remained a year after (next dry season, 12 months PI survey) ITS installation both for adult females (OR = 0.21, 95% CI 0.121–0.36, P = <0.001) and blood fed females (OR = 0.24, 95% CI 0.133–0.442, P = <0.001) (Table 1). Likewise, significant differences were observed on the total abundance of adult females (IRR = 0.12, 95% CI 0.061–0.249, P = <0.001) and blood fed females (IRR = 0.16, 95% CI 0.081–0.298, P<0.001) after the installation of ITS (wet season, 6 months PI survey) (Table 1). Significantly less indoor female *Ae. aegypti* (IRR = 0.19, 95% CI 0.114–0.309, P<0.001 and less blood fed females (IRR = 0.23, 95% CI 0.133–0.4, P<0.001) were still observed a year after the installation of ITS on the next dry season (Table 1).

### Impact of ITS on houses with pools of female *Aedes* positive for arbovirus

From 900 houses sampled during the study, 13% (117/900) were positive to *Ae. aegypti* females. A total of 139 *Aedes* female pools (mean of 1.2/ house positive to females), of which 74% were blood fed mosquitoes, were analysed for DEN/CHIK/ZIK virus diagnosis. A high number of pools, 108 pools (77.7%), were positive to ZIK virus indicating a strong signal of epidemic spread. All pools were negative to DEN/CHIK viruses. No significant differences were observed between study arms in the house positivity to ZIKV at baseline (OR = 0.6, 95% CI 0.07–6.32, P = 0.72) (Table 1). However, statistically significant differences were observed on the positivity for ZIK virus at the subsequent PI survey (OR = 0.15, 95% CI 0.08–0.29, P<0.001) during the rainy season. A year after the installation of ITS (dry season), these differences remained significant (OR = 0.24, 95% CI 0.15–0.39, P<0.001). The estimated intervention effectiveness in reducing ZIK infection, ITS$_{eff}$, was 85% (6 months) and 76% (12 months), or an average of 80.5%.

### Community acceptance and social perception on effectiveness

Three main reasons encouraged the participation of the residents from Juan Pablo II: the perception and worries about the high risk for *Aedes*-borne diseases transmission in the community (39%), the rationality and efficacy of the intervention in reducing mosquito-human contacts (25%), and that initially enrolled participants convinced more families through sharing their positive experiences about the effectiveness of the method (23%).

The installation process of ITS was considered very good for 91% of respondents. Overall, 100% of the participants perceived an efficacy on mosquito reduction; either with i) no mosquitoes inside some houses (58%) or ii) reduced number of mosquitoes (40%). In terms of the epidemiological association, most of the participants (91%) interviewed did not report any

**Table 1. Comparison of *Ae. aegypti* indoor-adult-based entomological indicators between treated (ITS) and untreated (control) groups at Juan Pablo II houses (n = 900) in Merida, Mexico.** Odds ratios (OR) and incidence rate ratios (IRR) with 95% confidence intervals are shown.

| Survey | Arms | Mean | SE (mean) | OR | P value | 95% C.I. |
|---|---|---|---|---|---|---|
| **House positive for *Aedes* females (Proportion)** | | | | | | |
| Baseline (Dry season 2016) | Control | 0.03 | 0.01 | 0.49 | 0.53 | 0.054–4.471 |
| | ITS | 0.01 | 0.01 | | | |
| 6 months PI (Rainy season 2017) | Control | 0.43 | 0.04 | 0.15 | <0.001* | 0.081–0.26 |
| | ITS | 0.10 | 0.02 | | | |
| 12 months PI (Dry season 2017) | Control | 0.17 | 0.03 | 0.21 | <0.001* | 0.121–0.36 |
| | ITS | 0.04 | 0.02 | | | |
| **Houses with Blood fed *Aedes* (Proportion)** | | | | | | |
| Baseline (Dry season 2016) | Control | 0.03 | 0.01 | 0.49 | 0.53 | 0.054–4.471 |
| | ITS | 0.01 | 0.01 | | | |
| 6 months PI (Rainy season 2017) | Control | 0.37 | 0.04 | 0.18 | <0.001* | 0.097–0.325 |
| | ITS | 0.09 | 0.02 | | | |
| 12 months PI (Dry season 2017) | Control | 0.15 | 0.03 | 0.24 | <0.001* | 0.133–0.442 |
| | ITS | 0.04 | 0.02 | | | |

| Survey | Arms | Mean | SE (mean) | IRR | P value | 95% C.I. |
|---|---|---|---|---|---|---|
| **_Aedes_ females per house (Average number)** | | | | | | |
| Baseline (Dry season 2016) | Control | 0.07 | 0.05 | 0.20 | 0.18 | 0.019–2.071 |
| | ITS | 0.01 | 0.01 | | | |
| 6 months PI (Rainy season 2017) | Control | 0.97 | 0.14 | 0.12 | <0.001* | 0.061–0.249 |
| | ITS | 0.12 | 0.03 | | | |
| 12 months PI (Dry season 2017) | Control | 0.21 | 0.04 | 0.19 | <0.001* | 0.114–0.309 |
| | ITS | 0.04 | 0.02 | | | |
| **Blood fed *Aedes* per house (Average number)** | | | | | | |
| Baseline (Dry season 2016) | Control | 0.06 | 0.04 | 0.22 | 0.20 | 0.022–2.247 |
| | ITS | 0.01 | 0.01 | | | |
| 6 months PI (Rainy season 2017) | Control | 0.68 | 0.10 | 0.16 | <0.001* | 0.081–0.298 |
| | ITS | 0.11 | 0.03 | | | |
| 12 months PI (Dry season 2017) | Control | 0.17 | 0.04 | 0.23 | <0.001* | 0.133–0.4 |
| | ITS | 0.04 | 0.02 | | | |

| Survey | Arms | Mean | SE (mean) | OR | P value | 95% C.I. |
|---|---|---|---|---|---|---|
| **House positive to female *Aedes* with arbovirus (ZIKV) infection (Proportion)** | | | | | | |
| Baseline (Dry season 2016) | Control | 0.02 | 0.01 | 0.66 | 0.720 | 0.069–6.318 |
| | ITS | 0.01 | 0.01 | | | |
| 6 months PI (Rainy season 2017) | Control | 0.36 | 0.04 | 0.15 | <0.001* | 0.081–0.295 |
| | ITS | 0.08 | 0.02 | | | |
| 12 months PI (Dry season 2017) | Control | 0.15 | 0.03 | 0.24 | <0.001* | 0.153–0.385 |
| | ITS | 0.04 | 0.02 | | | |

* Significant differences (P<0.05).

case of DEN/CHIK/ZIK virus infection within their families after the installation of mosquito screens on doors and windows. Interviewees did not acknowledge feeling any temperature increase attributable to the screening (77%); some reported a little increase on the temperature of the houses (19%) but related to specific day-hours such as mid-day. Finally, most of the participants (93%) said to be satisfied and recognised ITS as an effective method for the prevention of DEN/CHIK/ZIK transmission (96.43%). Families definitively recommended (100%) the scaling-up of the intervention, because the multiple positive outcomes perceived.

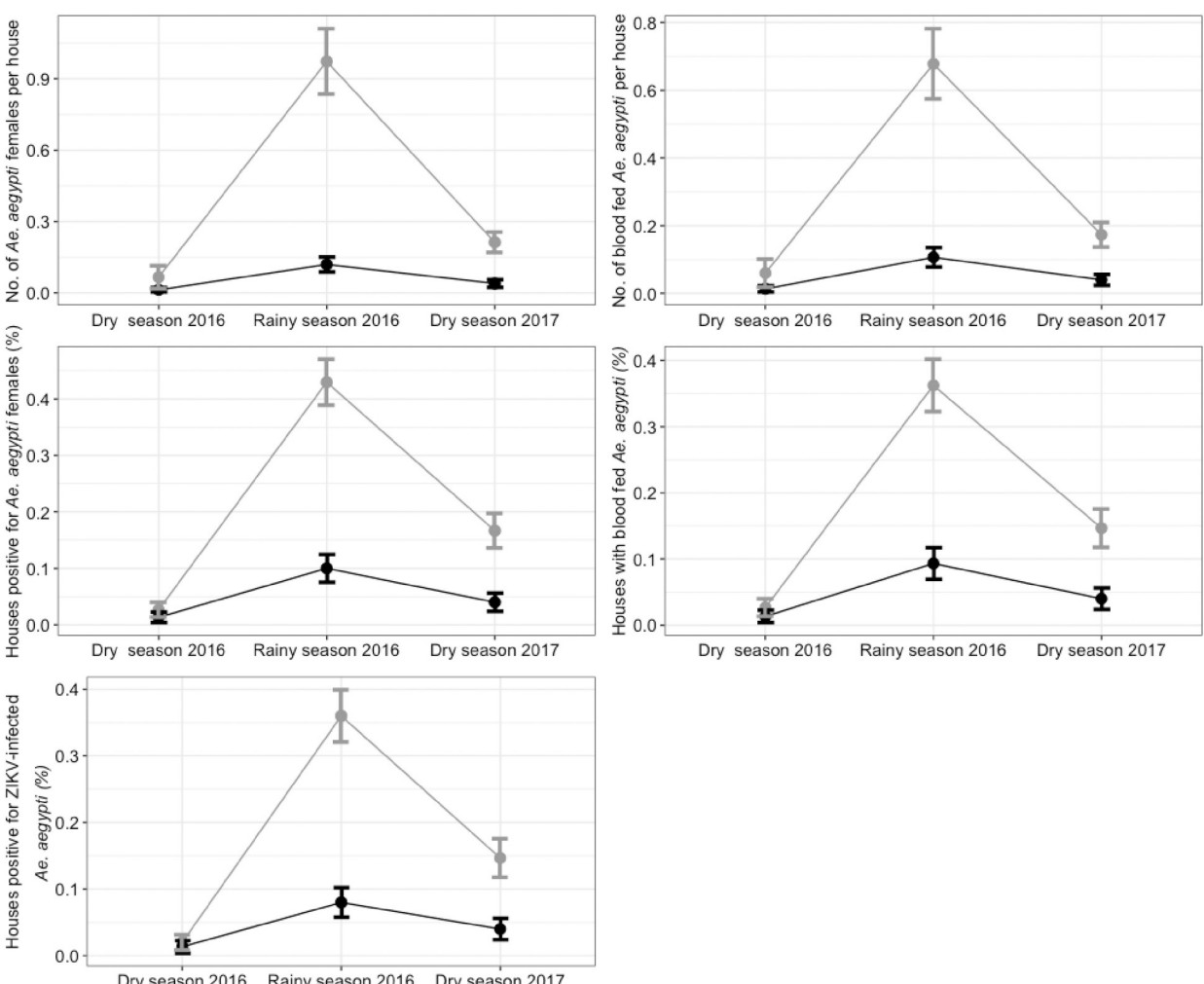

**Fig 2. Entomological indicators of impact.** Comparison between treated (black line) and untreated (gray line) arms of *Ae. aegypti* indoor adult based indicators for Merida, Mexico. The intervention (installation of ITS) was implemented between June-July 2016 (rainy season). Error bars show the standard error of the mean.

## Discussion

Screening entry-points of a house to prevent the access of adult mosquitoes -particularly *Aedes aegypti* females- is expected to decrease the number of vectors, human exposure to infective mosquito bites and therefore, reduce dengue, chikungunya and Zika transmission [1,2,15,31]. Here we provide evidence of the protective effect of ITS in reducing not only the entomological risk (presence and abundance of *Aedes* females and those blood-fed indoors), but also a reduction of an epidemiological proxy of the risk of transmission of ATDs (indoor *Aedes* females infected with ZIK virus). A house protected with ITS on doors and windows in this study at Merida, not only had ≈ 84% less chance of having *Ae. aegypti* females in comparison with a non-screened house during the peak of the mosquito season, but also and very importantly, had ≈ 80% less chance of having ZIK infected *Ae. aegypti* females inside in comparison with a non-screened house.

During 2016, a total of 10, 007 cases of dengue (2,064), chikungunya (69) and Zika (7,874) were recorded in Merida. Zika was first recorded in the city in 2016 and was the most frequent

*Aedes*-transmitted disease (78.4%) reported in Merida [32]. A cohort study conducted in pregnant woman at the same time of our study found ZIKV attack rates of ~30% [33], indicating high levels of virus transmission in Merida. Such high transmission was evidenced in the number of positive pools found within our study houses. Although this study was not designed to test the epidemiological impact of ITS at the human population level, results reported in the present study were in the context of a Zika outbreak, so they provide evidence that ITS/HS could give high protection against circulating arbovirus in mosquitoes, reducing significantly indoor *Aedes* presence, density and ZIKV infection. This also illustrates the importance of adult collections (and adult derived indicators including infection) rather than larval indicators for the evaluation of control methods targeting the adult stage.

Evidence of protection of ITS against local *Ae. aegypti* populations was observed despite the presence of a high level of resistance to pyrethroids in the local mosquito population. Previous studies from our research team have characterized the susceptibility/resistance levels of *Ae. aegypti* from Juan Pablo II (JP), with knockdown resistance (allele frequency of I1016 of 82% and C1534 of 93%) to pyrethroids [16,18]. Mosquito populations from JP are from moderately resistant to completely susceptible to alpha-cypermethrin, the active ingredient of the LLIN material used for the ITS intervention evaluated in this study. The insecticidal activity of ITS has shown to be effective (>70% mortality) as a chemical barrier, in addition to the benefits of house screening as a physical barrier [16,18], to exclude and kill mosquitoes and eventually protect against mosquito bites, which is epidemiologically relevant if most transmission occurs indoors.

ITS or HS have advantages over other approaches -as a preventive method- because once installed, they are permanently fitted, protect individuals and the whole family, require little additional work or behavioural change by household members, and are associated with high overall satisfaction and acceptance levels [20]. Some degree of success -and failure- has been reported after interventions with LLINs against dengue vectors (measured usually with immature based indicators) when used as curtains hanged on windows and doors [34–41]. However, unprotected doors and windows because peoples' habit of 'folding' the curtains during the heat of the day; compromise house mosquito-entry points [19,40]. Fixed screens covering permanently doors and windows eliminate this problem.

In the present study, ITS was very well accepted by the community, with a perceived efficacy on reductions on mosquito abundance and biting, and furthermore, reduction in other domestic insect pests; evidence that reinforces the positive outcomes found in other studies [20,42]. In the case of ITS, two main limiting factors for its accessibility by the community have been identified. Firstly, LLINs are not yet commercially available for public and/or in the retail market in Mexico, and secondly (also applicable for HS), the initial expenditure of the installation of aluminium framed-screens with high-quality materials is costly. Current implementation research from our group is focused on how to overtake these limitations to enhance community access to ITS or HS, including cost-saving strategies i.e. the use of less- expensive materials rather than aluminium frames, or with a Do-it-yourself strategy. Further implementation research is also exploring how much are the families are willing to pay and to find supplementary support by local governments or other funding schemes as part of a "safe housing" initiative or micro-credits.

While very encouraging, our findings have suffered from limitations in our study design and methods. The main limitation has been our lack of evidence of epidemiological impact of ITS on the human population. As an entomological trial, we never set to evaluate the impact on arbovirus infection in humans. However, our significant findings of a reduction in ZIKV infection in *Ae. aegypti* shed light into our inability to detect and test members of each house. Other limitations of the study include the limited number of houses, which prevented to

evaluate any population-level effect of ITS scale-up, as well as detailed information of the distribution of *Ae. aegypti* in the peridomicile. While in Merida *Ae. aegypti* is found primarily indoors [43], the lack of Prokopack aspirations outdoors limits our understanding of the true effect of ITS on *Ae. aegypti*. If both the chemical and mechanical barriers act on mosquitoes, one might expect to see a significant reduction in peridomestic *Ae. aegypti* indices; driven by reductions in adult density and their oviposition. Conversely, if only ITS acts as a mechanical barrier, one may expect not to see a dramatic reduction in vector densities if opportunities for human biting occur in the peridomicile. Either way, future studies should focus on the effect of ITS beyond the indoor environment and focus on spillover of mosquitoes to neighbouring houses (if any) as well as the potential long-term effect of ITS on the evolution of resistance to the active ingredient used in screens.

"Mosquito- proofing" of houses with house-screening has been a historic recommendation of environmental management [44] based on changes to human habitation to exclude vectors and reduce human-vector-pathogen contact. Mosquito-proofed housing and environmental management are recognised as part of the success in eliminating malaria in high-income countries [4,7,45,46]. A notable example is the construction of the Panama Canal, during which IVM was implemented as early as 1904, including the screening of living quarters and draining standing water, to reduce yellow fever and malaria [47]. Even tough, HS was largely ignored for policies & programs for the prevention and control of ATD; and it was not until the Zika emergency that the WHO [48], and their regional offices, finally emphasised the prevention and protection against mosquito bites using physical barriers such as window screens [49]. To complicate things further, and even nowadays, the evidence on the effectiveness of the current "toolbox" for ABDs is mixed in terms of "arboviral control" and not specific for Zika, mainly because the lack of scientific evidence (both insufficient to dengue and also because Zika was a newly emerged disease) [12,50].

There is an opportunity to demonstrate and support that HS can be a sustained protective barrier for families and the domestic environment as recommended by the World Health Organization [1,10,11]. HS (and/or housing improvement) should be "actively endorsed" and part of the current paradigms for urban vector-borne disease control [2]. Housing improvement is considered a public health intervention compatible with the integrated vector management strategy for *Ae. aegypti* in Mexico [51]. The strategy "safe housing and safe water" which consists of installing mosquito nets on doors and windows (either with or without insecticide) and keeping the patio clean and taking care of the stored water, is specifically recommended; nevertheless, it´s implementation by the vector control program of the Mexican MoH hasn't been accomplished yet. It is clear that housing improvements are far beyond of the budget of the MoH worldwide, and therefore, it is critical to involve other sectors, particularly the housing, urban planning and infrastructure sectors [10]. Protecting the home from *Ae. aegypti* and *Aedes*-transmitted diseases is a current and active challenge, with utmost relevance during COVID-19 (and post-COVID-19), due to strains in public health personnel, budgets, resources and vector control 'readiness'.

The results presented in this study further add to a growing body of evidence demonstrating that ITS/HS is a promising new paradigm for the control of *Ae. aegypti*, an antropophilic, endophilic, endophagic and day-biting species. The observed reduction in household *Ae. aegypti* infestation and importantly, on mosquito infection rates during a transmission period, could impact virus transmission in a measurable way, with evidence indicating good potential for sustainability, given the high levels of acceptance and popularity among targeted communities, and justify a second phase for larger trials (thousands of households) quantifying the effectiveness of ITS/HS on stronger epidemiological endpoints (human sero-conversion or infection).

We recently started the implementation of different high-quality, innovative interventions to complement traditional *Ae. aegypti* control in Merida, México, with a strong collaborative work with local authorities. The protection of houses with ITS received support from the local and national government It is under consideration how to expand *Aedes*-proof housing to as many homes as possible, conceivably as a targeted intervention for high-risk areas (hot-spots) and vulnerable populations of endemic localities.

## Supporting information

**S1 Table. Data supporting Table 1 and analysis of this article.**
(XLSX)

## Acknowledgments

The authors would like to express their sincere gratitude to the UCBE-UADY for the laboratory and insectary facilities for the development of this Project and to the residents of Merida for giving access to their homes for the collection of field material. The nets employed in this study were donated by the company Public Health Supply and Equipment de Mexico, S.A. de C.V.

## Author Contributions

**Conceptualization:** Pablo Manrique-Saide, Josué Herrera-Bojórquez, Norma Pavía-Ruz, Gonzalo Vazquez-Prokopec.

**Data curation:** Josué Herrera-Bojórquez, Emilio Trujillo-Peña, Josué Villegas-Chim, Ahmed M. M. Ahmed, Azael Che-Mendoza.

**Formal analysis:** Pablo Manrique-Saide, Josué Herrera-Bojórquez, Josué Villegas-Chim, Nina Valadez-González, Azael Che-Mendoza, Gonzalo Vazquez-Prokopec.

**Funding acquisition:** Pablo Manrique-Saide, Hugo Delfín-González, Norma Pavía-Ruz.

**Investigation:** Pablo Manrique-Saide, Josué Herrera-Bojórquez, Emilio Trujillo-Peña, Josué Villegas-Chim, Ahmed M. M. Ahmed, Jorge Palacio-Vargas, Norma Pavía-Ruz, Adriana E. Flores.

**Methodology:** Pablo Manrique-Saide, Josué Herrera-Bojórquez, Nina Valadez-González, Azael Che-Mendoza, Adriana E. Flores, Gonzalo Vazquez-Prokopec.

**Project administration:** Pablo Manrique-Saide, Anuar Medina-Barreiro, Hugo Delfín-González, Azael Che-Mendoza, Norma Pavía-Ruz.

**Resources:** Pablo Manrique-Saide, Anuar Medina-Barreiro, Hugo Delfín-González, Jorge Palacio-Vargas, Adriana E. Flores, Gonzalo Vazquez-Prokopec.

**Supervision:** Pablo Manrique-Saide, Josué Herrera-Bojórquez, Josué Villegas-Chim, Jorge Palacio-Vargas, Norma Pavía-Ruz.

**Writing – original draft:** Pablo Manrique-Saide, Josué Herrera-Bojórquez, Azael Che-Mendoza, Adriana E. Flores, Gonzalo Vazquez-Prokopec.

**Writing – review & editing:** Pablo Manrique-Saide, Azael Che-Mendoza, Gonzalo Vazquez-Prokopec.

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
