## [Decision Letter · Decision Letter 0]

28 Jul 2020

Dear Prof. Manrique Saide,

Thank you very much for submitting your manuscript "Insecticide-treated house screening protects against Zika-infected Aedes aegypti in Merida, Mexico" for consideration at PLOS Neglected Tropical Diseases. As with all papers reviewed by the journal, your manuscript was reviewed by members of the editorial board and by several independent reviewers. In light of the reviews (below this email), we would like to invite the resubmission of a significantly-revised version that takes into account the reviewers' comments. 

We cannot make any decision about publication until we have seen the revised manuscript and your response to the reviewers' comments. Your revised manuscript is also likely to be sent to reviewers for further evaluation.

Sincerely,

Pattamaporn Kittayapong, Ph.D.

Associate Editor

Ann Powers, Ph.D.

Deputy Editor

Reviewer's Responses to Questions

**Key Review Criteria Required for Acceptance?**

**Methods**

-Are the objectives of the study clearly articulated with a clear testable hypothesis stated?

-Is the study design appropriate to address the stated objectives?

-Is the population clearly described and appropriate for the hypothesis being tested?

-Is the sample size sufficient to ensure adequate power to address the hypothesis being tested?

-Were correct statistical analysis used to support conclusions?

-Are there concerns about ethical or regulatory requirements being met?

Reviewer #1: -The objectives of the study are clearly articulated and an important hypothesis was stated and tested.

-Study design: The study design was appropriate, but the total number of Aedes collected was relatively small. I find the differences convincing, but the represent relatively modest, albeit statistically significant differences. Again not many studies have demonstrated these kind of differences, linked to a proxy for transmission, but I might temper the enthusiasm a little. I would also commend the research team in taking advantage the introduction of ZIKV. Was the study planned prior to ZIKV introduction or were you able to react that quickly.

-No sample size calculation is included - this does not bother me, but trialists would be interested in if the study was powered for a particular effect size. If not I might describe how you got this trial up and running so quickly to take advantage of measuring ZIKV transmission the value this type of trial can contribute to evidence-base for vector-borne diseases.

-Statistics look appropriate to me, and Table 1 provides the "raw" numbers, again some care needs to be taken in talking about the significance.

-ZIKV testing - the authors note the "surprising number of positive pools for ZIKV. The lack of DENV/CHIKV provides some confidence, but my guess is the more laboratory inclined reviewers might be a bit nervous about this. You might provide a bit more discussion on this issue. If we could see the proportion of houses with Aedes aegypti that tested positive for ZIKV it would provide some additional perspective. Was the laboratory blinded to the cluster allocation?

Reviewer #2: The objective is clear and the methodology to achieve it is sound. 

Sample size calculation is not reported.

**Results**

-Does the analysis presented match the analysis plan?

-Are the results clearly and completely presented?

-Are the figures (Tables, Images) of sufficient quality for clarity?

Reviewer #1: -The article is simple and straight forward and the differences are apparent.

-Tables are good.

- Any insecticide resistance testing?

Reviewer #2: Results are correctly reported and presented. The quality of images is poor.

**Conclusions**

-Are the conclusions supported by the data presented?

-Are the limitations of analysis clearly described?

-Do the authors discuss how these data can be helpful to advance our understanding of the topic under study?

-Is public health relevance addressed?

Reviewer #1: -The conclusions are supported by the data presented.

-Limitations: I did not see this addressed directly and would have liked to see some discussion. Although the results speak for themselves, I feel the conclusions could be tempered a bit. This is 3 collection period (only 2 post intervention), it would have been far more convincing if monthly or bimonthly collections were done. They weren't and I'm sure there were economic constraints. While infected mosquitoes is an important proxy, it does not represent disease. Was any attempt made to look at MOH case reports or reports from study population (also flawed). Again this is incredibly promising but I would tone the language down in the abstract at least. 

-Excellent discussion of public health relevance, but I would encourage the authors to mention COVID-19 which makes this type of intervention even more relevant.

-One thing missing from the discussion, is a review (not necessary to be complete) of other studies using Insecticide treated materials, I believe there are some other studies from the Yucatan by another group and an RCT with less promising results using epi endpoints was recently published by Lenhart et al. in PLOS NTD. Again I think just tempers the conclusions, may not work everywhere in the same way.

Reviewer #2: Conclusions are supported by results. 

It is not reported if the sample size calculation has been performed so we do not know if this has affected the power of the analysis.

**Editorial and Data Presentation Modifications?**

Reviewer #1: The paper is well written and clear, but a some minor modifications addressing my previous comments would result in a better manuscript.

Reviewer #2: I strongly invite authors to add the numbers of mosquitoes and houses positive for Zika infected mosquitoes in Table 2 (it is just an additional column).

**Summary and General Comments**

Reviewer #1: The manuscript shows convincing evidence of an impact of Household screening on Aedes aegypti population parameters and Aedes infected mosquitoes. This is should be of high relevance, this type of data on an infection proxy is rare so this should be of high interest to PLOS NTD and the ATD vector control community.

General Comments

- The paper is clear and well done and makes a convincing argument for the value of HS.

- Study is of good size but not huge. Took advantage of the high transmission period associated with ZIKV, but there is no discussion of human transmission in the study area or houses. I understand the study may have not been able to test the population, but the lack of discussion of what sort of disease was seen feels a little like a read flag. ZIKV cases are talked about in the intro but very generally.

-There is not mention of insecticide resistance testing, it would have been important to include if done, and any context would have been useful. My understanding is pyrethroid resistance is high thorough out Mexico, so this issue at least deserves some discussion.

-A paragraph in the discussion about the potential limitations of the study would be useful.

-A paragraph in the discussion describing some of the previous work done on Insecticide treated curtains also be important.

-You might mention the importance of this intervention in light of the COVID-19 outbreak.

Reviewer #2: (No Response)

PLOS authors have the option to publish the peer review history of their article (what does this mean?). If published, this will include your full peer review and any attached files.

Reviewer #1: No

Reviewer #2: No
---

## [Decision Letter · Decision Letter 1]

20 Nov 2020

Dear Prof. Manrique Saide,

We are pleased to inform you that your manuscript 'Insecticide-treated house screening protects against Zika-infected Aedes aegypti in Merida, Mexico' has been provisionally accepted for publication in PLOS Neglected Tropical Diseases.

Best regards,

Pattamaporn Kittayapong, Ph.D.

Associate Editor

Ann Powers, Ph.D.

Deputy Editor

Reviewer's Responses to Questions

**Key Review Criteria Required for Acceptance?**

**Methods**

-Are the objectives of the study clearly articulated with a clear testable hypothesis stated?

-Is the study design appropriate to address the stated objectives?

-Is the population clearly described and appropriate for the hypothesis being tested?

-Is the sample size sufficient to ensure adequate power to address the hypothesis being tested?

-Were correct statistical analysis used to support conclusions?

-Are there concerns about ethical or regulatory requirements being met?

Reviewer #1: As this is a review of revised manuscript I will confirm that I had few concerns about the study methodology and the authors essentially addressed all of my "added" suggestions which were optional.

I'm completely satisfied here.

Reviewer #2: The study is well conceived, methods are appropriate and statistical analysis is adequate. Proper community engagement has been performed.

**Results**

-Does the analysis presented match the analysis plan?

-Are the results clearly and completely presented?

-Are the figures (Tables, Images) of sufficient quality for clarity?

Reviewer #1: Again all my mild concerns were addressed.

- one additional piece of information I would be interested in is a comparison between the blood-fed and non-blood fed pools of mosquitos, one could this was xenodiagnosis in a sense. Did you test the other species? This is not necessary here, but just something I failed to ask in the last revieve.

Reviewer #2: Data are analised according to the plan and the results and statistical analysis are clearly presented. In the PDF I have downloaded Figures are blurred.

**Conclusions**

-Are the conclusions supported by the data presented?

-Are the limitations of analysis clearly described?

-Do the authors discuss how these data can be helpful to advance our understanding of the topic under study?

-Is public health relevance addressed?

Reviewer #1: Affirmative to all the questions asked by PLOS NTD, and again I commend the authors for considering all of my previous suggestions.

Reviewer #2: The Discussion section is extensive and well thought. Results fully support it.

**Editorial and Data Presentation Modifications?**

Reviewer #1: No changes needed.

Reviewer #2: (No Response)

**Summary and General Comments**

Reviewer #1: This is ready to go.

Reviewer #2: The manuscript has been improved and I do not have additional comments. Congratulation to the authors for demonstrating the effectiveness of a control strategy against Aedes aegypti and the arboviruses it transmits.

PLOS authors have the option to publish the peer review history of their article (what does this mean?). If published, this will include your full peer review and any attached files.

Reviewer #1: No

Reviewer #2: No

---

## [Editor Report · Acceptance letter]

12 Jan 2021

Dear Prof. Manrique Saide,

We are delighted to inform you that your manuscript, "Insecticide-treated house screening protects against Zika-infected Aedes aegypti in Merida, Mexico," has been formally accepted for publication in PLOS Neglected Tropical Diseases.

Best regards,

Shaden Kamhawi

co-Editor-in-Chief

Paul Brindley

co-Editor-in-Chief
